

# Impact of urban canopy meteorological forcing on aerosol concentrations

Peter Huszar[1], Michal Belda[1], Jan Karlický[1], Tatsiana Bardachova[1], Tomas Halenka[1], and Petr Pisoft[1]

[1]Department of Atmospheric Physics, Faculty of Mathematics and Physics, Charles University, Prague, V Holešovičkách 2, 180 00 Prague 8, Czech Republic

**Correspondence:** P. Huszar (huszarpet@gmail.com)

**Abstract.** The regional climate model RegCM4 extended with the land-surface model CLM4.5 was coupled to the chemistry transport model CAMx to analyze the impact of urban meteorological forcing on the surface fine aerosol ($PM_{2.5}$) concentrations for summer conditions over the 2001-2005 period focusing on the area of Europe. Starting with the analysis of the meteorological modifications caused by urban canopy forcing we found significant increases of urban surface temperatures (up to 2-3 K), decrease of specific humidity (by up to 0.4-0.6 $gkg^{-1}$) reduction of wind speed (up to -1 $ms^{-1}$) and enhancement of vertical turbulent diffusion coefficient (up to 60-70 $m^2s^{-1}$).

These modifications translated into significant changes in surface aerosol concentrations that were calculated by "cascading" experimental approach. First, none of the urban meteorological effects were considered. Than, the temperature effect was added, than the humidity, the wind and finally, the enhanced turbulence was considered in the chemical runs. This facilitated the understanding of the underlying processes acting to modify urban aerosol concentrations. Moreover, we looked at the impact of the individual aerosol components as well. The urban induced temperature changes resulted in decreases of $PM_{2.5}$ by -1.5 to -2 $\mu gm^{-3}$, while decreased urban winds resulted in increases by 1-2 $\mu gm^{-3}$. The enhanced turbulence over urban areas results in decreases of $PM_{2.5}$ by -2 $\mu gm^{-3}$. The combined effect of all individual impact depends on the competition between the partial impacts and can reach up to -3 $\mu gm^{-3}$ for some cities, especially were the temperature impact was stronger in magnitude than the wind impact. The effect of changed humidity was found to be minor. The main contributor to the temperature impact is the modification of secondary inorganic aerosols, mainly nitrates, while the wind and turbulence impact is most pronounced in case of primary aerosol (primary black and organic carbon and other fine particle matter). The overall as well as individual impacts on secondary organic aerosol is very small with the increased turbulence acting as the main driver. The analysis of the vertical extend of the aerosol changes showed that the perturbations caused by urban canopy forcing, besides being large near the surface, have a secondary maximum for turbulence and wind impact over higher model levels, which is attributed to the vertical extend of the changes in turbulence over urban areas. The validation of model data with measurements showed good agreement and we could detect a clear model improvement at some areas when including the urban canopy meteorological effects in our chemistry simulations.



# 1 Introduction

Among many types of impacts of urban areas on environment the impact on the atmospheric environment is regarded to be the 'most important and most far-reaching' (Folberth et al., 2015). A major component of this impact is the direct influence of urban canopy on meteorological conditions: cities are largely covered by artificial surfaces and they affect the physical properties of the air above in a very specific way resulting in increases of temperatures, reductions of winds, increases turbulence and other meteorological modifications (Lee et al., 2011; Huszar et al., 2014, 2018; Karlický et al., 2018).

The most known aspect of urban influence on meteorological conditions is the formation of the urban heat island (UHI) which has been the subject of large number of studies since the early 80s (Oke, 1982). UHI and other related meteorological effects were studied by many authors and impacts on temperature, wind-speed, turbulence, structure of the boundary layer were identified (Basara et al., 2008; Gaffin et al., 2008; Roth, 2000; Kastner-Klein et al., 2001; Hou et al., 2013; Angevine et al., 2003). Cities further influence humidity, precipitation and the hydrological cycle in general (Richards, 2004; Rozoff et al., 2003). With the introduction of urban-canopy parameterizations and models of different complexity, modeling approaches describing the urban effects on meteorology and climate became widespread, examining both local (Wouters et al., 2013; Hou et al., 2013) and regionals scales (Feng et al., 2013; Trusilova et al., 2008; Struzewska and Kaminski, 2012; Huszar et al., 2014; Karlický et al., 2018). Huszar et al. (2014) showed that urbanization can contribute to regional warming (e.g. Huszar et al., 2014) and determine the climate of whole regions (Květoň and Žák, 2007), as the impact usually exceeds the geographical location of the city itself and propagates to larger scales.

Over urban areas, thus, the meteorological conditions are largely perturbed. Consequently, as air chemistry is strongly linked to meteorological conditions, it is expected that modifications in meteorological parameters will result in modifications in species concentrations as well, as it was already shown by many regarding the climate change related meteorological changes and their impact on air-quality (Huszar et al., 2011; Katragkou et al., 2011; Juda-Rezler et al., 2012).

In particular, the UHI triggered higher urban temperatures modify chemical reaction rates and particle nucleation and they influence also dry deposition velocities and wet scavenging rates (Seinfeld and Pandis, 1998). Further, e.g. Hidalgo et al. (2010) and Ryu et al. (2013b) showed that UHI can generate urban-breeze circulation resulting in pollutant transport from and to cities depending on the daytime but also on the surrounding orography and/or the presence of coasts (Ganbat et al., 2015; Li et al., 2017b). Urban surfaces however act in the opposite direction a well: higher drag induces wind stilling and thus suppresses of the dispersion of urban emissions and secondary pollutants into regional scales. Surface heterogeneities in urban areas further enhance turbulence and increased eddy-transport helps pollutant transport to upper layers of urban boundary layers (UBL; Stutz et al., 2004). In general, a very strong link is identified between the state of the UBL and pollution (Masson et al., 2008).

The above listed effects act however simultaneously in a rather complex manner requiring coupled modeling approaches. Most of the work done focused on gas-phase chemistry, especially ozone ($O_3$) and nitrogen oxides ($NO_x$) changes due to urban land-surface forcing. Martilli et al. (2003) and Sarrat et al. (2006) focused on one single city, Athens and Paris and found significant impact on pollutant concentrations, mainly due to changed turbulence when urban surfaces are considered. Civerolo et al. (2007), using a bulk approach for the description of urban surfaces, predicted considerable increase in episode-maximum





8-h average $O_3$ concentration due to future urbanization. Struzewska and Kaminski (2012) looked at southern Poland region and found reduction of primary pollutants ($NO_x$ and CO) due to enhanced vertical mixing. Recently, Fallmann et al. (2016) analyzed ozone changed after urban greening utilization and found $O_3$ reduction due to temperature mitigation, but increases of primary pollutant due to the associated decreased mixing. A large number of authors focused on Chinese cities and urban

areas (Wang et al., 2007, 2009; Xie et al., 2016a; Zhu et al., 2017). They calculated an ozone concentration increase due to urbanization which has a similar magnitude than due to future emission changes and changed climate, emphasizing the importance of considering urban canopy effects in air-quality modeling. Liao et al. (2014) investigated how different urban canopy parameterizations influence the air-quality prediction for a Chinese agglomeration. Sulfur dioxide ($SO_2$) changes due to urbanization were modeled by Chen et al. (2014) who found significant decreases of this important primary pollutant over

urban areas caused mainly by increased vertical mixing over urban canopy. Ryu et al. (2013a); Ryu et al. (2013b) modeled the urbanization impact on ozone concentrations over Seoul, Korea and identified strong urban-breeze circulation greatly affecting ozone concentrations, and, this was found to be further modulated by the effect of anthropogenic heat released from the city. Regarding aerosols, Zhu et al. (2017) investigated the impact of the change of land-use from natural to artificial due to urban expansion on coarse particle matter (PM10) and found decreases driven mainly by enhanced vertical eddy-transport. Increases

of PBL height and turbulence over urban areas were the main reason for decreases of primary pollutant concentration near the surface in Xie et al. (2016b) and Li et al. (2017a). Over Toulouse, France, Masson et al. (2008) analyzed the how fine mode aerosol interact with urban boundary layer and found strong role of the vertical dispersion in modulating its concentration. Recently, de la Paz et al. (2016) found that the application of urban canopy models instead simple "bulk" approaches have positive impact on model accuracy for both gaseous species ($NO_2$ and $O_3$) and $PM_{2.5}$. For fine aerosol they found that the

main driver for changes are the lower wind in urban area that are closer to observed values than those modeled with simple approaches or without considering urban surfaces at all. For Paris, France, Kim et al. (2015) showed that when using urban canopy model that triggers stronger vertical mixing in the model over urban areas, $PM_{2.5}$ concentrations are lower and agree better with observations. The important consequences landuse changes and, particularly, urbanization has on PM concentrations were noted by Tao et al. (2013) too.

The listed studies point to different meteorological changes occurring over urban areas that influence local and regional air pollution often in an opposite manner. Indeed, when evaluating the integrated chemical effect of the urban canopy meteorological forcing, one meet a number of difficulties. First of all, the individual meteorological components (like temperature changes, modifications of turbulence etc.) are often counteracting. Enhanced urban temperatures trigger higher reaction rates for production as well as chemical destruction of secondary pollutant. Decreases in wind speeds block pollutants to disperse

into larger scales, however this is true also for their precursors which stay close to sources and can trigger their destruction, like in case of ozone-NOx interaction. However, at the same time, enhanced vertical mixing contribute to their removal which decreases the concentrations. Another complexity in the urban meteorology/air chemistry interaction is brought by the fact that the urban induced meteorological features as well as emissions are not uniformly distributed in time and their daily cycles have specific features. Maximums and minimums in their peaks occur sometimes simultaneously leading to the amplifications

of the effects. In case of secondary aerosols, the situation is even more complicated as their concentrations are influenced not



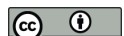

only directly via the urban meteorological effects, but also via changes in their gas-phase precursors. The overall impact on aerosol concentration is than the complex combination of the impact on individual aerosol components which, in case of secondary ones, is largely modulated by the impact on the source precursors. It is clear that integrated meteorology and air-quality modeling framework is an inevitable tool to analyze the changes of air pollutants due to urban canopy land-surface forcing in

detail.

Here, we introduce a model estimate of the impact of the urban canopy meteorological forcing on fine aerosol (PM2.5) concentrations over European urban areas. This study is a follow-up to Huszar et al. (2018) where the impact was analyzed from the perspective of gas-phase chemistry, focusing especially on NOx and ozone. Here, we make a further step and look at aerosols concentrations and as a novelty, our study, as one of the firsts, will investigate each component of $PM_{2.5}$, as it

is expected that different components of aerosol response differently to urban meteorological forcing. Another novelty of the study is that instead of modeling selected short periods (although interesting from meteorological perspective), we perform continuous long-term simulations in order to capture the long-term average impact. Our study is motivated by the appreciation that urban fine aerosol still represents a substantial threat to public in large cities (Cheng et al., 2016) and represent an important aspect of cities air pollution (Huszar et al., 2016a). In order to implement measures for its reduction, the proper knowledge

of the contributors to their levels is crucial and this has to include the potential contribution from land-use changes related to urbanization as well.

## 2   Experimental setup

### 2.1   Models

Models used in this study were introduced and used with a very similar setup in Huszar et al. (2016a, b) and Huszar et al.

(2018). Here we provide a rather brief but self-consistent description of them. The regional climate model RegCM (version 4.4) was used as a meteorological driver (Giorgi et al., 2012). For convection and large-scale precipitation, the Grell and SUBEX scheme were invoked, respectively (Grell, 1993; Pal et al., 2000). The planetary boundary layer processes were modeled with the Holtslag scheme Holtslag et al. (1990). Radiative transfer calculation were conducted by the NCAR Community Climate Model Version 3 (CCM3; Kiehl et al., 1996).

The Community Land Model version 4.5 (CLM4.5; Lawrence et al., 2011; Oleson et al., 2013) was chosen to describe the land-cover processes. CLM4.5 provides a more comprehensive description of land surface processes compared to the simple BATS land-surface model which is originally included in the RegCM model Dickinson et al. (1993). CLM4.5 further contains the CLMU urban canopy scheme (Oleson et al., 2008) based on the traditional canyon representation of urban areas. The canyon consists of roofs, walls, and canyon floor. Trapping of solar and long-wave radiation within the canyon is taken into

account. Momentum fluxes are calculated for the urban landunit using a roughness lengths and displacement height typical for the canyon. Anthropogenic heat from air conditioning and heating is computed online within CLMU.

We have chosen to use the CLM4.5 scheme as it produces stronger and more realistic UHI compared to BATS scheme and gives more emphasized wind speed decrease (see Huszar et al., 2014, 2018). It further reduces the overestimation of evaporation





in summer seen in the BATS scheme and hence models precipitation with a higher accuracy, which was already concluded by Wang et al. (2015) who compared these two schemes.

For chemical simulations, the chemistry transport model (CTM) CAMx version 6.30 ENVIRON (2016) was coupled offline to the regional climate model. CAMx is an Eulerian photochemical CTM implementing multiple gas phase chemistry mechanism options (CBV, CB6, SAPRC07TC). In this study, the CBV scheme (Yarwood et al., 2005) is invoked. CAMx further implements static two mode as well as multi-sectional particle size treatments, wet and dry deposition of gases and particles and calculates the composition and phase state of the ammonia-sulfate-nitrate-chloride-sodium-water inorganic aerosol system in equilibrium with gas phase precursors with the ISORROPIA thermodynamic equilibrium model (Nenes and Pandis, 1998) activated in our setup. For aerosol, we invoked the two mode particle size treatment and the semi-volatile equilibrium scheme called SOAP (Strader et al., 1999) for the formation of secondary organic aerosol (SOA) was used.

As an offline couple, no feedbacks from the simulated species concentration changes on RegCM radiation/microphysical processes (cloud/rain) were taken into account. Huszar et al. (2016b) showed minor effects of these feedbacks in long-term averages so we consider this as a reasonable simplification. The RegCM generated meteorological fields are translated into CAMx input using the RegCM2CAMx preprocessor (Huszar et al., 2012). In RegCM2CAMx, we replaced the O'Brien (1970) method for calculating the vertical eddy diffusion coefficients (required by CAMx) with the more advanced Byun (1999) scheme. This leads to better match of model results with measurements (Eben et al., 2005). With this choice, an inconsistency is brought to the modeling set-up as the methods calculating vertical diffusion coefficient differ between the climate and chemistry models. To achieve the highest degree of coupling between the driving climate model and a chemistry transport model, the Holtslag et al. (1990) scheme produced vertical diffusion (Kv) parameters should have been used directly to drive the vertical diffusion in CAMx. As RegCM4 does not support to output these parameters (in the used versions and configurations), technically much simpler was to take the vertical profiles of wind speed and temperature as well as the PBL height from the driving model and apply a diagnostic method to calculate the Kv for CAMx (provided by Byun (1999)). Applying a "non-consistent" method in calculating Kv for CTMs however does not implicate less accurate results than directly coupling the PBL parameters, as showed by Lee et al. (2011).

### 2.1.1 Data and simulations

The models were applied on the same domain and configuration as in Huszar et al. (2018), this means 10 km × 10 km horizontal resolution centered over Prague, Czechia with 160 x 120 x 23 gridboxes in x, y, and z direction. It is clear that with such horizontal resolution the fine scale structure of chemical transformation of emissions cities cannot be resolved. With this resolution, emissions from cities enter the atmosphere trough a few grid boxes while considering instant dilution on the model scale. It is clear that due to the non-linearity of chemical processes this can lead to some errors in the final species concentrations. This effect is considered, however, to be small in case of city emissions (Markakis et al., 2015). Many previous studies also agreed that resolutions similar to ours are suitable to model the regional impact of urban emissions Varghese et al. (2011); Tie et al. (2010) although some overestimation of secondary species is documented when using coarser model resolution (Karlický et al., 2017).




The top model levels correspond to 50 hPa for the climate model. CAMx was set up only on the lowermost 18 levels (up to approximately 10 km). Lateral boundary conditions for the regional climate calculations were taken from a 50 km x 50 km resolution experiment carried out with the RegCM4 model within the EURO-CORDEX initiative (Vautard et al., 2013) driven by the ERA-Interim reanalysis (Simmons et al., 2010).

The TNO emissions prepared in the framework of the FP7 MEGAPOLI project were used (Kuenen et al., 2010) as anthropogenic emissions source. They provide high resolution (1/8° longitude 1/16° latitude, roughly 7 km x 7 km) European data of annual emission estimates for $NO_x$, $SO_2$, non-methane volatile organic compounds (NMVOC), methane ($CH_4$, ammonia ($NH_3$), carbon monoxide (CO) and particulates (PM10 and $PM_{2.5}$) in ten activity sectors. For each sector, specific temporal disaggregation factors and NMVOC speciation profiles were applied to decompose the annual sums into hourly emissions

following to Winiwarter and Zueger (1996). For chemical initial and boundary conditions (ICBC), a 30 km x 30 km domain run was performed covering whole Europe. This large domain run was, in turn, driven by time-space invariant chemical ICBC. This choice of chemical ICBC resulted in some model biases listed in Huszar et al. (2016a), especially regarding ozone. Meteorology-dependent biogenic emissions of isoprene and monoterpenes (BVOC) were considered in the study following Guenther et al. (1993).

Landuse was extracted from the USGS data while urban landunit percentage was derived from the $0.05° \times 0.05°$ resolution LandScan2004 dataset based on census, night-time lights satellite observations and road proximity (Jackson et al., 2010). They include further urban canyon parameters and surface physical properties. For chemical simulations, the landuse was kept the same for all experiments in order to separate the effect of meteorological changes only.

    The experiments cover a period 2001-2005 with the year 2000 as a spinup. Summer months (JJA) are analyzed in our study

when urban effects are the most pronounced. With the regional climate model, one pair of experiments was carried out: (i) a reference simulation (NOURBAN experiment) which does not considers urban landunit, meaning that urban gridboxes are replaced with the one that is most typical for the surrounding gridboxes, most often crops; (ii) and a simulation that considers urban surfaces and parameterized with the urban canopy model CLMU (URBAN experiment).

    Within the impact of the simulated meteorological changes on chemistry, the following effects are considered of 1) mod-

ified temperature (t–impact); 2) modified absolute humidity (q–impact); 3) modified wind field (uv–impact) and 4) modified turbulence (via changes in the vertical eddy diffusion coefficient; kv–impact). A number of experiments with the chemistry model CAMx were carried out depending of which effect is ex-/included. The reference CAMx simulation is driven by the reference climate simulation and is denoted in the same way: NOURBAN. After, a "full" experiment was carried out where all the listed effects are considered (the URB_t+q+uv+kv experiment). To obtain a more detailed picture about the role each

process plays, additional experiments were carried out by turning on the individual components of the overall meteorological effect one-by-one following the methodology in Huszar et al. (2018). Accordingly, the URB_t experiment considered only the urban temperature effects. In the URB_t+q experiment, both temperature and humidity effects are accounted for. The effect modified wind was added to the URB_t+q+uv experiment. Finally, in the "full" experiment the effect of modified eddy diffusion coefficients on chemistry was considered. With such cascading approach one can analyze the separate impact of individual

meteorological parameters and their contribution to the total impact. Although this experimental approach must lead to some



inconsistency in the meteorological driving fields, e.g. the vertical layer structure (layer interface heights) defined in the CAMx input will be not consistent with the increased temperature in the URB_t experiment, these simulations serve only to explain how the chemical changes are "building up" from "building up" of the meteorological influences and an assumption is made according to which the possible effect of these inconsistencies is small when averaged over a long period. Further, the role

of the ordering of individual impacts was investigated. Short runs were carried out for summer 2001 (not shown here) where the temperature, humidity, wind and turbulence changes are considered in different order and the results for individual components were practically the same. The effect of temperature driven BVOC changes is not accounted for as it is rather negligible (Huszar et al., 2018).

In Huszar et al. (2016a), an extensive validation is provided for the chemistry simulated by the RegCM4-CAMx couple

for the 2001-2010 period. Here, the same configuration and input data is used except that the CLM4.5 surface model is used instead of the older BATS scheme. We expect only a minor impact on the simulated species concentrations as lateral boundary conditions and emissions were identified as the main sources of model bias and these are the same in this study. We only provide a spatial comparison of modeled $PM_{2.5}$ concentrations with observation (see further), mainly to examine possible model improvement when urban canopy effects are considered in the driving meteorology. Regarding the meteorology, we use

the same pair of (URBAN-NOURBAN) experiments to drive CAMx as in Huszar et al. (2018). They compared temperature and precipitation fields with observational data and found an underestimation of modeled temperatures by 2-3 K and both under and overestimation of precipitation by +/- 30 % depending on the location. In this study, we make the assumption that the simulated impact of urban canopy on climate is influenced by these biases only in a minor way, as the impact is calculated always as difference and thus the biases partly are eliminated.

## 3   Results

### 3.1   Impact on meteorological conditions

Huszar et al. (2018) provided both the spatial impact of the urban canopy on meteorological conditions and the diurnal cycle of the impact over selected cities. We here limit our presentation to the diurnal cycles extended with the impact on the absolute humidity. If not specified otherwise, the diurnal cycles are plotted in sundial time (approximately UTC+1h for almost the entire

domain). We analyzed a large number of cities, however the results showed a high degree of similarity over each of them. Therefor, here we present only two cities, Berlin and Prague, respectively. For the humidity, we also show the spatial impact presented for day and night-time, considering 11–4 pm and 0–5 am, respectively. Shaded areas mean statistically significant difference on the 98% level using t–test.

Fig. 1, the JJA average diurnal cycle of (from left to right) the near surface temperature, humidity, 10 m wind speed and

vertical eddy diffusion coefficient over selected cities (solid lines) and their vicinities (dashed lines) is plotted. The pattern regarding the temperature impact is very similar for both chosen city. As expected, the experiment with urban surfaces predicts higher temperatures over urban areas, while the difference reaches almost zero during morning hours. Maximum temperatures are enhanced by about 0.5 to 1 °C when urban landuse type is considered. The largest impact occurs during late afternoon and





evening peaking around 8 pm. The plots further reveal almost negligible impact over the city vicinities (increase up to 0.2 °C). The diurnal temperature range is reduced by 1.5 to 2 °C.

The diurnal cycle of absolute humidity is plotted in Fig. 1 2nd column. In absolute values, maximum values are reached during late afternoon hours, when the evaporation from the surface is highest, reaching 9–10 $\mathrm{gkg^{-1}}$. As expected, humidity

is decreased due to the presence of urban surfaces, caused mainly due to decreased evaporation and increased runoff. The decrease is largest simultaneously with the largest absolute values occurring during 19–20 pm. It can reach -0.7 to -0.8 $\mathrm{gkg^{-1}}$. The plot also reveals that during nighttime, the urban impact on absolute humidity can be positive. Similarly, the humidity values from urban vicinity are slightly lower than over the city (seen in the URBAN experiment). This is probably connected to higher capacity of urban air to hold water vapor (due to higher nighttime temperatures), or in other words, rural air cools

more quickly during night and more vapor is removed by condensation on the surface. Higher humidity during nighttime due to urban surfaces is also seen on the spatial distribution of the impact in Fig. 2 where it can reach 0.2 $\mathrm{gkg^{-1}}$ especially for cities over the western part of the domain, probably due to more humid maritime climate they have. On the other hand, during daytime, a clear decrease of absolute moisture is modeled with peaks over cities often exceeding -1 $\mathrm{gkg^{-1}}$ (e.g. for Milan, or Budapest).

In case of the impact on wind, as expected, the wind speeds during the day are higher compared to those during night. The decreases can reach about -0.6 to -0.8 $\mathrm{ms^{-1}}$ and their timing match the timing of the highest absolute wind speeds during daytime. This is expected as the change is proportional to the absolute values. The impact on wind is smallest around 7 –8 pm, when the evening PBL transition occurs (Huszar et al., 2018).

The last column of Fig. 1 shows the diurnal cycle of the vertical maximum of the turbulent diffusion coefficient (Kvmax).

A systematic increase by 10-20 $\mathrm{m^2s^{-1}}$ is modeled due to urban surfaces throughout the day with maximum occurring during morning hours. The minimum is modeled during afternoon to late evening hours. The impact over the cities vicinity practically negligible.

## 3.2 Impact on aerosols

Regarding the impact of urban canopy meteorological forcing on aerosol concentration, we will start with the impact on the

$\mathrm{PM_{2.5}}$ surface concentrations. Next, the impact on its components, i.e. primary and secondary (in)organic aerosols, will be evaluated. For all aerosol type, besides the total impact, results will be provided also for the individual components (t-, q- ,uv- and kv-impact) as well. These aerosol components are considered: sulfates (PSO4), nitrates (PNO3), ammonium (PNH4), primary organic aerosol (POA), primary elemental carbon (PEC), other fine particulate matter (FPRM) and secondary organic aerosol (SOA). Further, for each components, both the spatial distribution of the impact and also the diurnal cycle over selected

cities is presented. In the spatial distribution figures, shaded areas represent statistically significant differences on the 98% level using t–test.



### 3.2.1   PM2.5

Fig. 3 shows the modeled 2001-2005 average surface concentration of $PM_{2.5}$ for the NOURBAN experiment (left) and for the experiment with all meteorological effects included ("URB_t+q+uv+kv"; right). Colored dots denote measured averages extracted from the European Environment Agency Airbase data background stations. Largest $PM_{2.5}$ values are modeled over

highly populated large agglomerations like Ruhr area in Germany, Po valley in norther Italy or souther Poland, where summer average values can reach 30-50 $\mu gm^{-3}$. The comparison to measured data shows some model underestimation, especially over central European countries and northern Italy, where model values ale lower by 5-10 $\mu gm^{-3}$. On the other hand, over large parts of Germany, model results agree with measurements or are slightly positively biased. If considering the urban canopy meteorological effects, $PM_{2.5}$ is clearly decreased by about 3 $\mu gm^{-3}$ over urban areas. This leads to better model results over

parts of western Europe, but over areas, where $PM_{2.5}$ is underestimated, the model bias increased.

The perturbation of $PM_{2.5}$ surface concentrations due to urban canopy meteorological changes is plotted in Fig. 4 for the temperature, humidity, wind and turbulence impact as well as the combined total impact (from left to right from top to bottom). Regarding the temperature impact, there is domain wide decrease peaking over cities with values from -1.5 to -2 $\mu gm^{-3}$, especially over western part of the domain (e.g. over the Ruhr area in Germany). Smaller decreases are modeled for central

and eastern European cities (up to -0.5 $\mu gm^{-3}$ decrease). The impact is seen not only over urban areas but propagates to rural ones as well, with statistically significant decreases around -0.1 to -0.2 $\mu gm^{-3}$. Changes in moisture content (decrease in average) cause only a slight decrease of $PM_{2.5}$ over most of the western part of the domain up to -0.1 $\mu gm^{-3}$. Over other parts, small increases are modeled. The urban induced wind stilling causes $PM_{2.5}$ to increase over most of the domain with peaks, as expected, over cities up to 2 $\mu gm^{-3}$. Statistically significant increases are modeled even over rural areas with less urban fraction

often exceeding 0.5 $\mu gm^{-3}$. $PM_{2.5}$ responses to the increased vertical mixing due to urban surfaces with decreases over cities by up to -2 $\mu gm^{-3}$ again mainly over the western part of the region in focus. Above some rural areas, statistically significant increases are modeled, however these are very small not exceeding 0.1 $\mu gm^{-3}$ and most often below 0.05 $\mu gm^{-3}$. The total impact of all the considered components show spatial similarities with the turbulence impact – it is enhanced or suppressed depending on the competition between the temperature (decreases) and wind impact (increases). However, over most of the

cities, the temperature impact is stronger in magnitude than the wind impact causing the total impact to be enhanced, often reaching -3 $\mu gm^{-3}$ decrease.

As it was already pointed out, the urban meteorological phenomenon have a well defined diurnal cycle. Thus, it can be expected that the above presented impacts on aerosol is not uniformly distributed across the day but will have a specific diurnal cycle too. In Fig. 5 the diurnal variation of the $PM_{2.5}$ absolute surface concentrations and their change over two representative

cities, Berlin and Prague is presented. The impact over other cities is qualitatively same in shape with only different magnitudes. The absolute values (solid lines) are plotted for the NOURBAN case and for the experiment with all urban meteorological effects considered ("URB_t+uv+q+kv"). The changes due to individual and combined urban meteorological effects are plotted with dashed lines. Fine particulates concentrations for the selected cities have a clear diurnal cycle with maxima (minima) around 18-22 (10-14) $\mu gm^{-3}$ occurring around 19-20 (13-14) pm. Regarding the diurnal cycle of the effect of individual





meteorological components there are however substantial differences. Temperature (orange) causes decreases by up to 1.5-2 $\mu$gm$^{-3}$ with maximum during nighttime, while the minimum occurs around noon (around -0.5 $\mu$gm$^{-3}$ change). As already seen in the spatial figure, the changes due to modified humidity (aquamarine) are very low not exceeding 0.05 $\mu$gm$^{-3}$. Wind decrease causes increase of urban PM$_{2.5}$ concentrations (green) by around 0.5-1 $\mu$gm$^{-3}$ during all the day, while changes are

higher when the absolute values are higher too, i.e. during evening and nighttime. A very clear daily cycle is modeled for the turbulence induced concentration change (olive) with a profound maximum (of the absolute change) occurring at 18-19 pm with values around -4 and -1.6 $\mu$gm$^{-3}$ for Berlin and Prague, respectively. The daily cycle of the total impact (purple) has a similar shape than the turbulence impact with maximum occurring around 18-19 pm reaching -5 and -2 $\mu$gm$^{-3}$ for Berlin and Budapest, respectively. A smaller secondary peak occurs during nighttime, when the temperature component is strong. In

summary, the most emphasized change is caused by increased turbulence, thus the total impact is dominated by this component too.

It is clear, that, in general, different components of the fine aerosol will contribute differently to the absolute concentrations as well as to the changes presented above. To, see, what is the relative importance of each aerosol type, we plotted in Fig. 6 the relative composition of the average summer aerosol for six selected cities. Largest contribution is made by sulfates and nitrates

being around 50 %, while sulfates dominate especially over eastern European cities. Ammonium constitutes about 15% of the total fine aerosol. Primary organic aerosol, primary elemental carbon (black carbon) and other fine particulate matter makes around 10, 5 and 10 %. Secondary organic aerosol has the lowest contribution around a few percents. In further, the impact of each component of the urban meteorological forcing on the individual aerosol type will be analyzed, starting with the primary aerosols. Note, that we ignored the humidity effect, as its contribution the the total PM$_{2.5}$ change turned out to be minor.

### 3.2.2 Primary aerosols

7 shows the spatial distribution of the urban meteorological forcing on the surface concentration of primary aerosol: POA, PEC and FPRM. In the chosen model configuration, these primary aerosol do no interact with chemistry and their emission as well as contribution to the total fine aerosol over urban areas is comparable (see Fig. 6) there for their response to different meteorological modifications is similar. The impact of temperature changes are minor, reaching only a few -0.01 $\mu$gm$^{-3}$ change

25   over urban areas. A more emphasized impact is modeled due to urban wind changes reaching 0.3-0.4 $\mu$gm$^{-3}$ over many cities. Increased turbulence leads to statistically significant changes almost over the entire domain with decreases up to -0.5 $\mu$gm$^{-3}$. The total impact reflects the opposite sign of the wind and turbulence changes and is somewhat lower reaching -0.3 $\mu$gm$^{-3}$ but over some areas where the wind impact dominates, it can be even positive up to 0.2 $\mu$gm$^{-3}$. Due to lower values the statistical significant changes occupy smaller areas limited mainly around large cities.

30   The diurnal cycle of the absolute urban primary aerosol concentration as well as changes due to urban meteorological forcing is shown in Fig. 8. The pattern is very similar for both selected cities with two maxima for the absolute values caused mainly by the diurnal cycle of urban emissions (mainly morning and evening heavy traffic). As seen in the spatial figures, the temperature impact is almost zero with a tiny negative peak during evening hours when the absolute values are highest. The wind induced changes are similar in magnitude all over the day with a maximum again during evening hours reaching 0.1-0.2 $\mu$gm$^{-3}$ for the



selected cities. The strongest impact is modeled for the turbulence effects with a very well expressed peak occurring during evening hours (around 18 pm). The strong kv-impact determines the cycle of the total impact as well, which is slightly smaller due to the positive impact of the wind changes. These results are similar to the diurnal cycle of $PM_{2.5}$ values expect that the t-impact is much stronger in case of $PM_{2.5}$. This is gives us a hint that the reason for this difference will lie probably in the contribution of secondary aerosol. In further, we will look at their contribution to the over final aerosol modifications.

### 3.2.3 Secondary aerosols

Fig. 9 plots the spatial distribution of the urban meteorological forcing on the surface concentration of secondary inorganic aerosol: PSO4, PNO3 and PNH4 as well as of secondary organic aerosol. The urban temperature changes cause decrease of each aerosol type, while the strongest decrease is modeled for nitrates, up to -1 $\mu gm^{-3}$. Sulfates decrease over urban areas by up to 0.3 $\mu gm^{-3}$ and ammonium is decreased by up to 0.4-0.5 $\mu gm^{-3}$. The smallest decrease is modeled for the SOA. Consequently, from secondary aerosols, nitrates contribute the most to the temperature induced $PM_{2.5}$ changes. In case of the wind-impact, there is an evident increase of concentration due to suppressed dilution from sources for all the examined aerosol component. Here again, the strongest increases are modeled for nitrates, up to 0.6-0.8 $\mu gm^{-3}$ (especially over western Europe), while for sulfates and ammonium, the change reaches 0.3-0.4 $\mu gm^{-3}$. For SOA, being a relatively minor modeled aerosol component, the wind induced changes encompass very small increases up to 0.015 $\mu gm^{-3}$. A more complicated response of surface aerosol concentrations is modeled for changes in the vertical eddy diffusion. The western part of the domain and over large cities, increase of the diffusion coefficient leads to decrease of surface aerosols up to -0.4 $\mu gm^{-3}$, especially for sulfates and ammonium. Over eastern Europe, a slight increase of secondary aerosol is modeled reaching 0.05 $\mu gm^{-3}$ over larger areas. The SOA is an exception here, where concentrations are suppressed all over the domain peaking over urban areas up to -0.04 $\mu gm^{-3}$.

As it combines both decreases (temperature- and turbulence impact) and increases (wind- and partly the turbulence impact), the total impact of all meteorological effects often tends to be smaller and the areas of statistical changes are occupying a smaller fraction of the domain. This is well seen in case of PSO4, where due to the combined effects of the urban meteorological forcing, statistically significant effects are modeled only for the Ruhr area in northwestern Germany, controlled by the temperature and turbulence induced decrease reaching -0.5 $\mu gm^{-3}$. For PNO3 the combined meteorological impact manifests itself as a statistically significant decrease of concentrations over large cities while the highest values (up to -0.8 $\mu gm^{-3}$) are modeled over the largest cities. A similar picture is obtained for ammonium, where statistically significant decreases are modeled for the western part of the domain. In each case above, statistically significant effects are controlled by the combined effect of temperature and turbulence, while the wind induced increase is too small to counterbalance them. For SOA, the situation is similar: the total impact is dominated by decreases due to temperature and turbulence induced decreases, and reaches -0.04 $\mu gm^{-3}$ decrease over urban areas.

In order to better understand the modeled response of secondary inorganic aerosol to changed meteorological forcing, we plot the response of the precursor species as well. In Huszar et al. (2018, Fig. 6), the response of NOx to urban induced meteorological changes was presented and increases of species surface concentrations were modeled due to decreased wind




up to 2 ppbv. Due to turbulence increase, NOx responded with decrease up to -4 ppbv. Here, in Fig. 10, the response of sulfur dioxide ($SO_2$) and ammonia ($NH_3$) surface concentrations are presented as precursors of sulfates and particulate ammonium (ammonium sulfates and ammonium nitrates). For both aerosol type, surface concentrations decrease due to increased urban temperatures, by up to -0.2 to -0.5 ppbv, and this is probably connected to increased dry deposition. Lower urban winds cause

reduced horizontal dilution resulting in higher concentrations: this is seen for both $SO_2$ and $NH_3$ that their concentrations is usually enhanced around cities by yo to 2-4 ppbv. However, in case of sulfur dioxide, some rural areas encounter decreases, probably due to the fact that transport is suppressed to this areas due to reduced wind-speeds. The same holds for ammonia, however the decrease here is very small. Enhanced urban turbulence leads to a clear decrease of $SO_2$ surface concentrations over and near urban areas, by up to -4 ppbv. In caser of $NH_3$, this decrease reaches -1 to -1.5 ppbv. A more diverse picture

is obtained for the combined effect of urban meteorological changes and both decreases and increases are encountered over and around cities between -3 and 3 ppbv (somewhat smaller values for ammonia). This is probably caused by the competing effect of wind caused increase and decrease due to increased vertical eddy transport. In summary, primary precursors respond to increased urban temperatures, decreased urban winds and increased turbulence by a slight decrease, larger decrease and increase, respectively.

The diurnal cycle of the impact of urban meteorological forcing on secondary (in)organic aerosol is shown in Fig. 11 for two selected cities, Berlin and Prague. As the formation of these, beginning with uptake in water phase and nucleation, is highly temperature dependent and lower temperature favor gas-to-particle partitioning, the absolute values (taken from the the NOURBAN and URB_t+q+uv+kv runs) are lower during daytime, as expected. This behavior is evident from the t-impact (orange): it follows the cycle of the urban impact on temperature being larges during nighttime hours when UHI is the strongest.

It is well seen that the temperature impact is the dominant impact for nitrates and ammonium (note the total–impact curve in purple). The changes of aerosol concentrations due to modified moisture content is negligible, as already seen for $PM_{2.5}$. The impact of wind changes follows the spatial result seen in Fig. 9 and characterized with increases: it is usually lower during daytime, when absolute values are also low compared to nighttime values.

The impact of increased turbulence is evident in case of SOA, where it seems to be the main contributor to the total-impact

for both cities. It causes only a small changes for nitrates and ammonium. In case of PSO4 however, the two cities differ: while in Berlin, increased vertical mixing removes some sulfates from the surface layer reducing its concentrations, for Prague, the behavior is more complicated and concentrations can even increase due to higher vertical mixing. The change is relatively small, however, in case of Prague and the concentrations can vary as a function of some secondary non-linear effects like transport from higher model levels to lower ones due to residual turbulence during nighttime.

In order to obtain an idea of the vertical extent these effects take place, we plotted, in Fig. 12 the vertical profile of individual impacts on $PM_{2.5}$ concentrations for Berlin and Prague. As expected, the temperature, turbulence and the total impact reduces surface concentrations while the wind increases them. Around 900-1200 m the changes quickly become very small, for the temperature effect almost negligible. At higher levels, the turbulence impact changes sign and shows a strong maximum around 2700 m. The same secondary maximum occurs for the wind impact. Consequently, the combined impact has this large positive

maximum too reaching 0.2-0.3 $\mu gm^{-3}$.



## 4  Discussion and conclusions

The modeled meteorological response due to introduction of urban canopy shows expected features. The impact on temperature and its daily cycle has a very similar magnitude as well as shape as in numerous previous studies examining European cities (Pichierri et al., 2012; Giannaros et al., 2013; Struzewska and Kaminski, 2012; Sarrat et al., 2006) but similar diurnal variations of UHI (which is close to the impact of urban land-surface on temperature) where obtained for cities over other continents (Gaffin et al., 2008; Ryu et al., 2013a). Due to notable increase of nighttime temperature and only a slight enhancement of the daytime ones, a strong decrease of diurnal temperature range is modeled, with values compared to previous work (e.g. Trusilova et al., 2008). The simulated wind speed changes are also consistent with previous model experiments preformed over Europe (Struzewska and Kaminski, 2012; Vautard et al., 2013) or over Chinese urban areas (Hou et al., 2013; Zhu et al., 2017). The sudden drop of wind impact to almost zero during evening hours is caused by relatively high downward momentum flux due enhanced turbulence (Huszar et al., 2018) during the evening transition period (Lapworth, 2003). Concerning the urban impact on the turbulence, our simulated eddy diffusion coefficient and their changes are in line with what was previously modeled over urban areas (Kim et al., 2015) or in general over complex terrain (ENVIRON, 2011), although they are slightly smaller. This is probably caused by the inconsistencies brought in by diagnostic calculation of vertical diffusivity values rather than directly taking them from the PBL scheme of the used driving model.

Due to urban canopy, humidity decreased in average in our simulations which is in line with expectations. Urban areas are covered with materials that have a low evapotranspiration and are covered with vegetation in minor. Further due to high run-off, the precipitated water is transported by sewer system away from urban areas. In summary, sources of moisture over urban areas are limited compared to rural surfaces with much higher fraction of vegetation (Richards, 2004). However, during nighttime a slight increase of the absolute humidity is observed in model. In general, this can be a result of both increased source of moisture and decreased sink. The first possible reason can be dismissed. Regarding the reduced sinks of moisture, this is a straight consequence of high urban temperatures. Compared to rural environments, urban areas can hold larger quantity of moisture as a consequence of the Clausius-Clapeyron equation. In rural areas during night, temperature drops often under dew-point resulting in condensation which act as a sink. This does not occur however at such extend in urban areas, where a reduced dew deposition has been observed in correspondence with the expectations (Richards and Oke, 2002). Hence, urban nighttime absolute humidity can be slightly higher, than the rural one, at least near the surface.

It has been shown, that the urban canopy meteorological forcing decreases fine aerosol concentrations and this decrease is strongest during late afternoon hours. The component analysis of the response revealed that the most important contributor to this change is the enhanced turbulence over cities which facilitates the removal of both the aerosol itself, but also of its precursors (as seen for $SO_2$, $NH_3$ and noted for $NO_2$). The diurnal change caused by all the considered meteorological changes has almost identical shape as the change caused purely by the turbulence enhancement, especially at high absolute values. Our results confirm the strong connection between aerosol concentration and vertical turbulent transport that has been confirmed already by many authors (e.g. Kim et al., 2015; Zhu et al., 2017). These last mentioned revealed, in line with our results





that urbanization triggered enhanced turbulence reduces the modeled aerosol concentrations. It is mainly the primary aerosol (organics, black carbon and other fine particulate matter) which has a most pronounced responses to the turbulence changes.

Regarding the wind impact, our expectation has been confirmed – there is a clear relationship between urban wind speeds and air pollution: reduced wind causes reduction in pollutant dispersion leading to increased concentration near the sources or vice versa. This is in line with the recent findings of Jandaghian and Akbari (2018) who showed, that albedo induced increases in urban wind-speeds result in decrease of $PM_{2.5}$ concentrations. The wind induced aerosol increase has a clear diurnal cycle that reflects the cycle of the absolute values: about two times higher increase is modeled for nighttime (from early evening to early morning) than for daytime.

The simulated changes due to urban temperature increase are evident only in case of secondary aerosols. This is expected as temperature strongly controls the gas-particle partitioning and nucleation of aerosols with high temperatures reducing the tendency to form particles. In this regard, it is not surprising, that temperature induced changes are strong in case of secondary aerosols and are negligible for primary ones. To support our finding about the temperature driven decrease of secondary aerosol concentrations, Juda-Rezler et al. (2012) and Dawson et al. (2007) earlier arrived to conclusion that secondary inorganic aerosol and SOA included in PM decrease as they shift from the particle to the gas phase with increasing temperature. In our simulations, this is most evident for nitrates, in line with Myhre et al. (2006) and Dawson et al. (2007).

Almost negligible impact on $PM_{2.5}$ is modeled due to changes in moisture content. It has a slight effect on dry deposition velocities of gases via increasing stomatal and cuticle resistance due to higher relative humidity (Zhang et al., 2003). Further, reduced moisture negatively affect the hydroxyl radical concentrations reducing the oxidative capacity of the air above urban areas. However, both processes are probably of minor importance as the moisture variations are small.

The simulated impact on secondary organic aerosol are in line with the expectation, although they are very small in magnitude. Higher urban temperatures negatively affect the nucleation rates of organic vapors leading the reduced SOA concentrations. Decreased urban winds reduce the dispersion of SOA leading to slight increase of condensations. Thirdly, enhanced urban turbulence favor removal of both SOA and its precursors (semivolatile hydrocarbons) resulting in lower concentrations, which seems to dominate the overall urban impact.

Our results showed that the effects of urban canopy meteorological forcing are getting weaker with increasing height, which is not surprising, as the meteorological changes are often limited to the first few model layers (Huszar et al., 2014). The turbulence here is a different case, and Huszar et al. (2018) showed that it can be significantly perturbed over cities at higher model layers as well (up to 2200 m). This can explain the vertical secondary maximum occurring for the turbulence impact: it transports pollutants from the surface to higher model layers, however, at certain height the turbulence is no longer affected by the urban areas and aerosol can thus accumulate, increasing the upper level concentrations. For the wind impact, the situation can be similar. Higher surface concentrations results in more PM transported to higher model layers.

An important question is whether the inclusion of urban canopy effects in the driving meteorological fields reduces the simulated aerosol bias. This is not clear from the results. Areas where fine aerosol was overestimated by the model, especially over western Europe, decrease of $PM_{2.5}$ reduced the model bias. On the other hand, over eastern Europe, the model error is enhanced as here aerosol was under-predicted in our model. In general, the effort of more realistic representation of different





processes in models does not imply always better model performance. However, this should not discourage to improve models and in our case, it is evident, that accounting for the urban induced meteorological changes in air quality simulations leads to modified concentrations that cannot be disregarded in future urban air quality oriented studies. Especially the temperature, wind and eddy diffusion fields must be correct and reflect urban conditions.

5    There are some further documented effects of urban canopy on meteorology. For example the urban breeze circulation (Hidalgo et al., 2010) which affects pollutant transport over city scales (Ryu et al., 2013a). Another meteorological phenomenon occurring over urban areas is the impact on convection: the UHI triggered convergence zone over urban areas favor the development of convection and increase the frequency of extreme precipitation (Zhong et al., 2017): directly influencing the air quality by decreased solar radiation (due to cloud cover) and increased wet removal of pollutants. These effects
10   however act over scales that were not resolved in our modeling setup thus applying higher resolutions for investigating the urban/meteorology/air-quality interactions is essential in future research.

*Data availability.*  All data modeled and analyzed in this study is available upon request. Please contact peter.huszar@mff.cuni.cz.

*Competing interests.*  The authors declare that no competing interests are present.

*Acknowledgements.*  This work has been funded by the project of OP-PPR (Operation Program Prague – Pole of Growth)
15   CZ.07.1.02/0.0/0.0/16_040/0000383 "URBI PRAGENSI - Urbanization of weather forecast, air quality prediction and climate scenarios for Prague", by project PROGRES Q47/Q16 – Programmes of Charles University, by the project SVV 2018 of the Charles University and by the project UNCE 2040202013. We further acknowledge the TNO MEGAPOLI emissions dataset from the EU-FP7 project MEGAPOLI (http://megapoli.info) and the providers of the AirBase European Air Quality data (http://www.eea.europa.eu/data-and-maps/data/aqereporting-1).



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





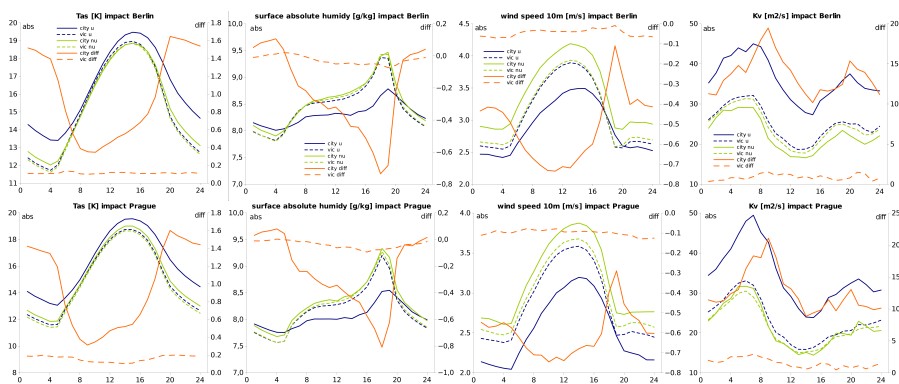

**Figure 1.** Diurnal cycle of the modeled near surface temperature in $^{\circ}$C, absolute humidity in $\mathrm{gkg}^{-1}$, 10 meter wind speed in $\mathrm{ms}^{-1}$ and the eddy diffusion coefficient at the 5th model level (about 700-800 m) in $\mathrm{m}^2\mathrm{s}^{-1}$ in local time for two selected cities, Berlin and Prague: blue - URBAN experiment, green - NOURBAN experiment; red - difference of URBAN and NOURBAN simulations; solid line - city averaged values, dashed line - city vicinity.

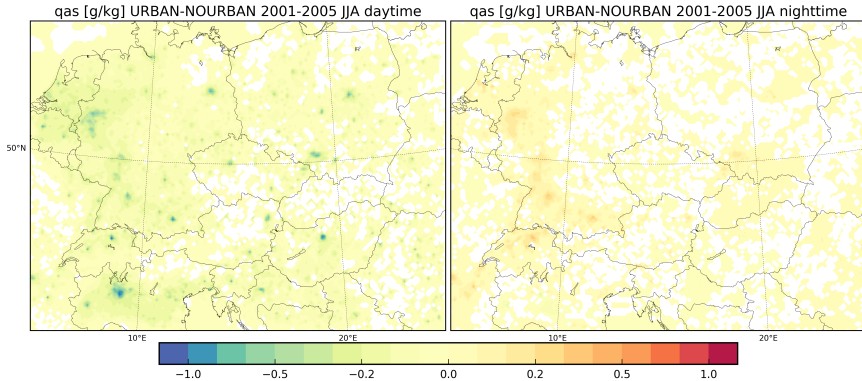

**Figure 2.** Impact of urban canopy on the near surface absolute humidity for 2001-2005 JJA in $\mathrm{gkg}^{-1}$ for daytime (left) and night-time (right). Shaded areas represent statistically significant impact on the 98% level using t-test.



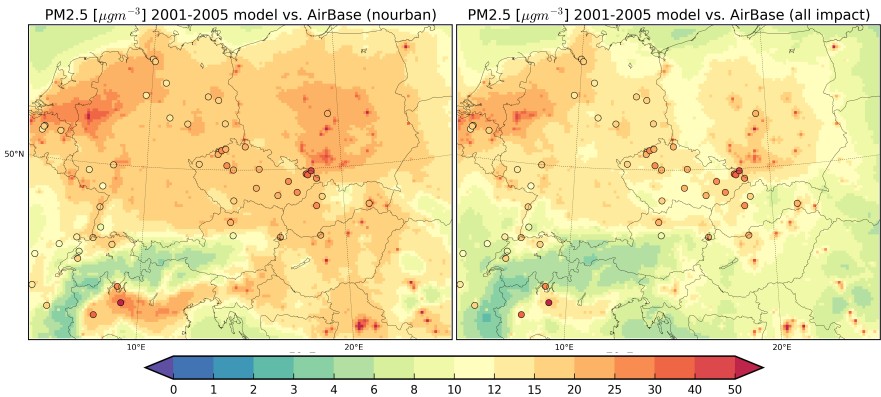

**Figure 3.** Absolute surface $PM_{2.5}$ concentrations in μgm$^{-3}$ averaged over the 2001-2005 JJA period for the NOURBAN experiment (i.e. without urban surface meteorological effects) and with the experiment where all urban meteorological effects are considered. Circles represent measured values from the Airbase database.

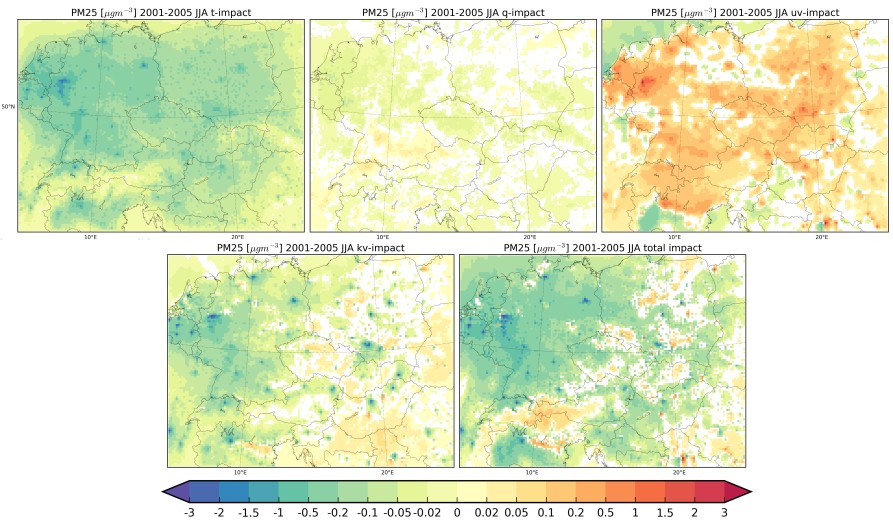

**Figure 4.** Impact of urban canopy meteorological forcing on $PM_{2.5}$ surface concentrations in μgm$^{-3}$ for the 2001-2005 JJA period for the (from left to right, up to down) temperature (t), humidity (q), wind (uv), turbulence (kv) and total impact. Shaded areas represent statistically significant impact on the 98% level using t-test.





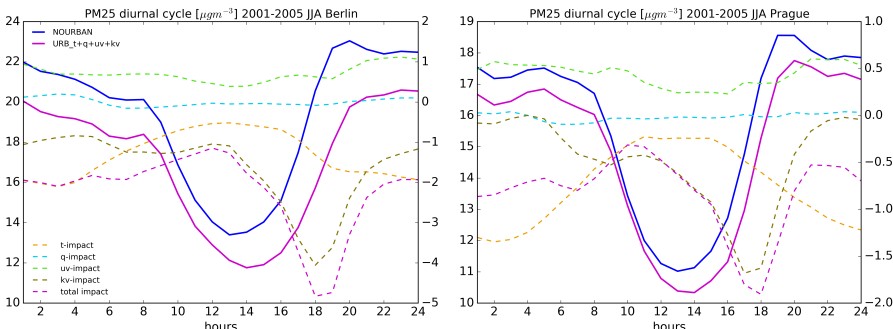

**Figure 5.** Impact of urban canopy meteorological forcing on $PM_{2.5}$ average diurnal cycle of surface concentrations for the 2001-2005 JJA period for two selected cities (Berlin and Prague) in $\mu gm^{-3}$. Bold lines represent the absolute concentrations (left y-axis) for the NOURBAN run (blue) and the total-impact "URB_t+q+uv+kv" run (purple). Dashed lines (right y-axis) show the change due to changes of individual meteorological components (temperature – orange, humidity – aquamarine, wind – green, turbulence – olive and total impact – purple).

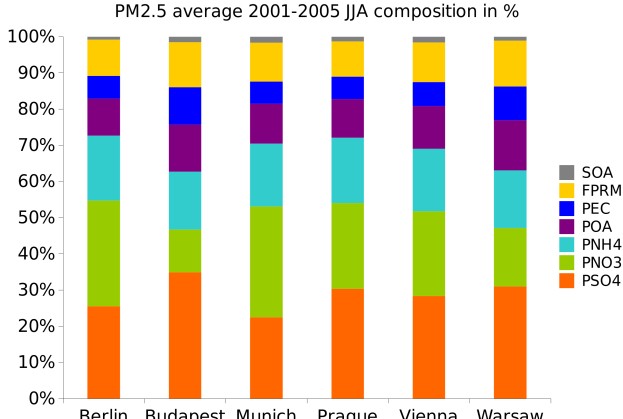

**Figure 6.** Average percentual composition of urban 2001-2005 JJA average $PM_{2.5}$ aerosol for selected cities: PSO4 - sulfates, PNO3 - nitrates, PNH4 - ammonium, POA - primary organic carbon, PEC - black carbon, FPRM - other fine particle matter, SOA - secondary organic aerosol



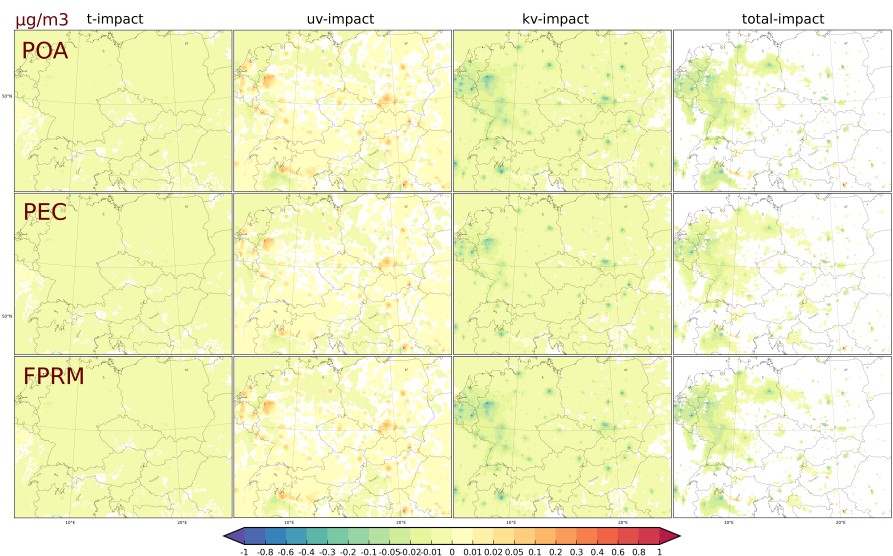

**Figure 7.** Impact of urban canopy meteorological forcing on (from top to bottom) POA, PEC, FPRM average 2001-2005 JJA surface concentrations in μgm$^{-3}$. The columns from left to right correspond to temperature, wind, turbulence and the total impact. Shaded areas represent statistically significant impact on the 98% level using t-test.





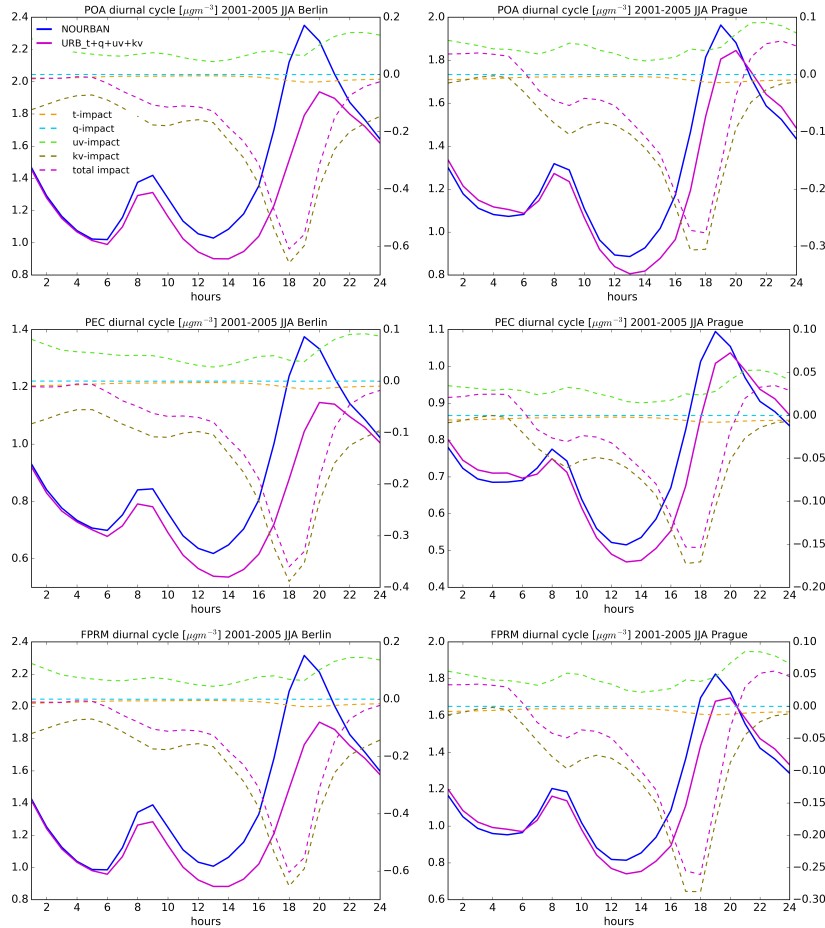

**Figure 8.** Impact of urban canopy meteorological forcing on (from top to bottom) POA, PEC and FPRM average diurnal cycle of surface concentrations for the 2001-2005 JJA period for two selected cities (Berlin and Prague) in µgm$^{-3}$. Bold lines represent the absolute concentrations (left y-axis) for the NOURBAN run (blue) and the total-impact "URB_t+q+uv+kv" run (purple). Dashed lines (right y-axis) show the change due to changes of individual meteorological components (temperature – orange, humidity – aquamarine, wind – green, turbulence – olive and total impact – purple).





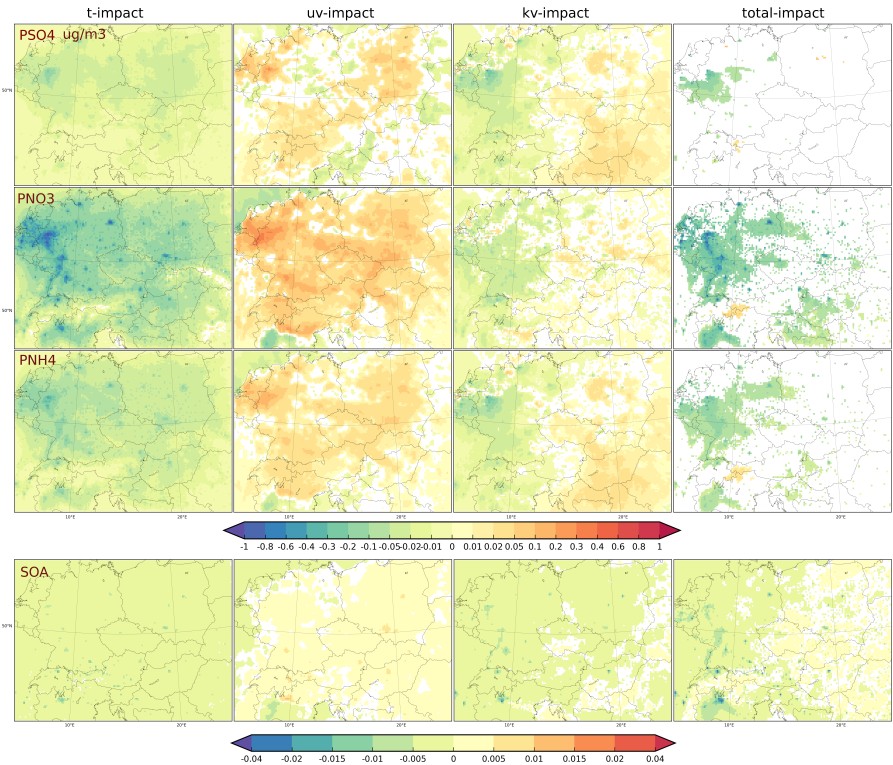

**Figure 9.** Impact of urban canopy meteorological forcing on (from top to bottom) PSO4, PNO3, PNH4 and SOA average 2001-2005 JJA surface concentrations in $\mu\mathrm{gm}^{-3}$. The columns from left to right correspond to temperature, wind, turbulence and the total impact. Shaded areas represent statistically significant impact on the 98% level using t-test.

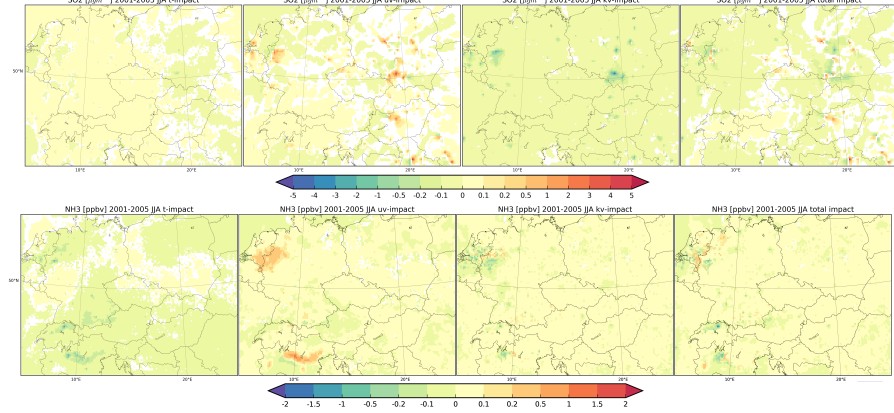

**Figure 10.** Same as Fig. 9 but for $SO_2$ (upper row) and $NH_3$ (lower row) in ppbv.




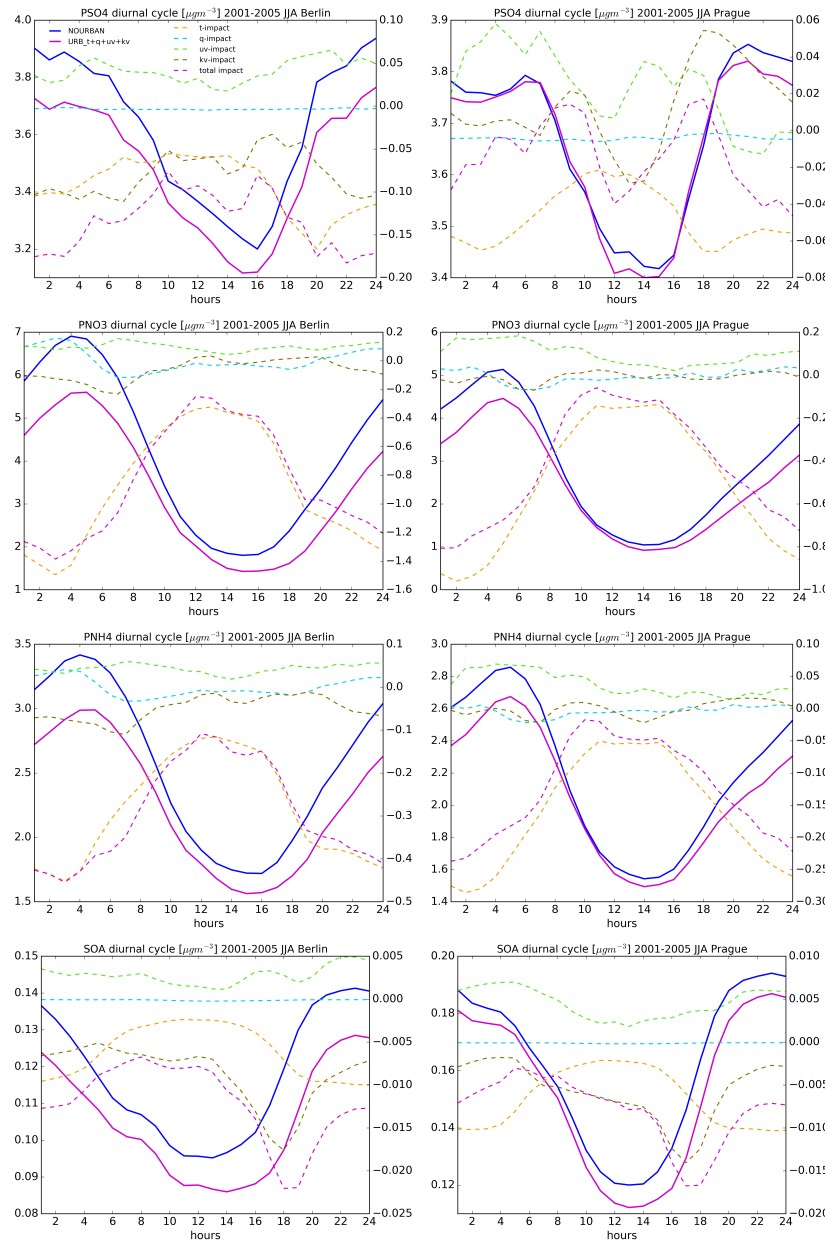

**Figure 11.** Impact of urban canopy meteorological forcing on (from top to bottom) PSO4, PNO3, PNH4 and SOA average diurnal cycle of surface concentrations for the 2001-2005 JJA period for two selected cities (Berlin and Prague) in $\mu gm^{-3}$. Bold lines represent the absolute concentrations (left y-axis) for the NOURBAN run (blue) and the total-impact "URB_t+q+uv+kv" run (purple). Dashed lines (right y-axis) show the change due to changes of individual meteorological components (temperature – orange, humidity – aquamarine, wind – green, turbulence – olive and total impact – purple).




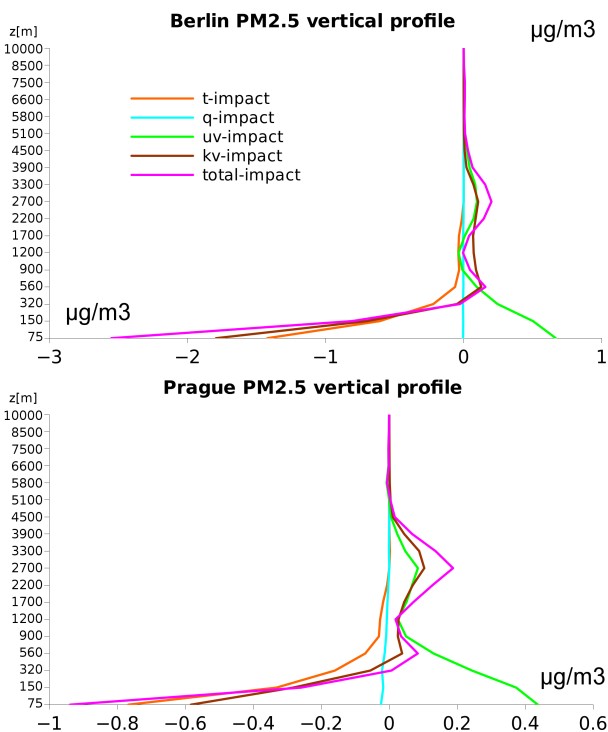

**Figure 12.** Impact of urban canopy meteorological forcing on the vertical profile of $PM_{2.5}$ concentrations for two selected cities, Berlin and Prague in $\mu gm^{-3}$ averaged over the 2001-2005 JJA period: temperature impact – orange, humidity impact – aquamarine, wind impact – green, turbulence – brown and total impact – purple). Vertical axis shows the layer interface heights in meters for the 18 CTM layers.