# Peer review of "Impact of urban canopy meteorological forcing on aerosol concentrations"

_Atmospheric Chemistry and Physics, 2018_

## Referee Comment (RC1) · Anonymous Referee #2 · 16 Jul 2018

General comments: This is an interesting paper investigating the role of urban canopy meteorological forcing on both aerosol as well as aerosol compounds' concentrations being a novelty of that study. The simulations have been performed for Europe for summer conditions over the period of 2001-2005 by the use of regional climate model RegCM4 coupled to the chemical transport model CAMx. The paper is well-written, clearly presents results and conclusions. It has considerable scientific significance. I have following comments before the paper is accepted to be published in ACP.

Specific comment: The CLM4.5 surface model included in the calculations is based on "the canyon representation of urban areas, described by building height and street width". These are quite detailed input data and also their impact is important in the lower scale than 10 km x 10 km used in the model simulations, I suppose also that

the availability of such data is rather limited. Please give the source of the data as well as discuss their representativeness in the 10 km resolution. The same questions (source of data, representativeness) arise from the way how "anthropogenic heat from air conditioning and heating" is calculated on-line in CLM4.5.

Technical comments: p. 2, l. 14: "regional" instead of "regionals";

p. 3, l. 16: "how" instead of "the how";

p. 4, l. 27: should be (Dickinson et al., 1993);

p. 5, l. 32-33: please give the references in parentheses;

p. 6, l. 7: should be (CH4);

p.7, l. 26: should be "Therefore". In this sentence please remove "respectively";

p. 7, l. 29: should be "In Fig. 1";

p. 7, l. 31: should be "for both cities";

p. 8, l. 6: please use "7-8 pm" instead of "19-20 pm". Please correct throughout the MS;

p. 8, l. 26: should be "all aerosol types";

p. 8, l. 27: please remove "as well";

p. 8, l. 29: should be "for each component";

p. 9, l. 5: should be "Po Valley in Northern Italy or in Southern Poland";

p. 9, l. 7: should be "Northern Italy". Please correct throughout the MS;

p. 9, l. 10: "is increasing" instead of "increased";

p. 9, l. 13: "of wide" instead of wide";

p. 10, l. 17: please add "respectively" in the end of the sentence;

p. 10, l. 21: "Fig. 7";

p. 10, l. 21-23: please clarify the sentence starting with "In the chosen...";

p. 12, l. 34-35: should be "...has also this large positive maximum, reaching...";

p. 14, l. 33: "In areas..." instead of "Areas...".

---

## Referee Comment (RC2) · Anonymous Referee #3 · 11 Sep 2018

General Comments: The manuscript by Huszar et al. is an interesting paper in which the authors combine the regional climate model RegCM4, land-surface model CLM4.5 and the chemistry transport model CAMx to investigate the impact of urban meteorological forcing on the surface fine aerosol (PM2.5) concentrations and its components during the summer seasons from 2001-2005 in Europe. The manuscript is well-written and the results and conclusions are properly presented. Considering its scientific significance, I recommend this manuscript to be accepted and published in ACP only after authors address the following comments.

Specific Comments: In P10, L15 authors mentioned that "largest contribution is made by sulfates and nitrates being around 50 %, while sulfates dominate especially over eastern European cities". What is the reason for the sulfate dominance over eastern

Europe?

It is not clear to me how the input data for building heights and street width is used in CLM4.5 on a 10 km × 10 km scale. What are the sources of this data?

In P11, L19 authors commented "The SOA is an exception here, where concentrations are suppressed all over the domain peaking over urban areas up to -0.04 $\mu$gm$-3$". What is the reason for this exception?

Technical Comments: Please correct all the typing errors throughout the manuscript. I have listed some of them below: P1, L4: "increase" instead of "increases"

P1, L8-9: "Then" instead of "Than"

P1, L11: "decrease" instead of "decreases"

P5, L32-33: References in parentheses?

P6, L7: "CH4"

P7, L26: "Therefore" instead of "Therefor"

P8, L6: "7-8 pm" instead of "19-20 pm"

P9, L5: "northern" instead of "norther"; "southern" instead of "souther"

P9, L7: "are" instead of "ale"

P10, L23: "therefore" instead of "there for"
* * *

---

## Author Comment (AC1) · 11 Sep 2018

Author response to the Referee #2's comments on manuscript "Impact of urban canopy meteorological forcing on aerosol concentrations" - acp-2018-415 by Peter Huszar et al.

We would like to thank to Referee #2 for the reviewing our manuscript and for all the comments, corrections and suggestions. We will consider all of them and our point-by-point responses follow.

Referee #2 comments:

Specific comment: The CLM4.5 surface model included in the calculations is based on "the canyon representation of urban areas, described by building height and street

width". These are quite detailed input data and also their impact is important in the lower scale than 10 km x 10 km used in the model simulations, I suppose also that the availability of such data is rather limited. Please give the source of the data as well as discuss their representativeness in the 10 km resolution. The same questions (source of data, representativeness) arise from the way how "anthropogenic heat from air conditioning and heating" is calculated on-line in CLM4.5.

Authors response: As written in the manuscript, the urban morphology parameters are obtained from LandScan2004 global 2D data (Jackson et al., 2010) which defines 132 regional categories (the world is divided into 33 regions with similarities in urban characteristics and each category is subdivided into 4 subcategories representing different urban intensities - tall building district (TBD), high density (HD), medium density (MD), and low density (LD)). For each bottom category, average building heights (H), urban canyon height-to-width ratios (H:W), and fraction of pervious surface (e.g., vegetation), roof area, and impervious surfaces (e.g., roads and sidewalks) are defined, among other parameters. Jackson et al. provide all of the data sources from which these data were compiled. We checked the data for particular cities over the domain and they are within the range of the typical urban geometry represented by central European cities (see Huszar et al., 2014 for a few values representative for Prague, Czech Republic). Urban landunit within CLM4.5 is represented as fraction in percentages of three (of the four in Jackson et al.) urban intensity (HD, MD and LD). This gives a reasonable description of urban coverage even at 10 km resolution and even small cities well below 10 km in diameter are accounted for. Of course, within the model gridbox and within one urban intensity, urban parameters do not vary in space, however we consider this variation within the uncertainty range of other inputs like boundary conditions or physical parameterization etc. Regarding the anthropogenic heat release, it is calculated from the heat conduction equation with a interior boundary conditions represented by interior temperature of the building. To this anthropogenic heat flux, another heat flux is added that accounts for the waste heat from air heating/conditioning. It is parameterized directly from the amount of energy required to keep the internal

building temperature between a prescribed maximum and minimum values, assuming 50% efficiency of the heating/cooling systems, see Oleson et al. (2008) for detailed description.

Changes in the manuscript: We included some more detailed description (in Section 2.1.1) of how the urban parameters are obtained for the region in focus including their representativeness using chosen resolution.

Technical comments: p. 2, l. 14: "regional" instead of "regionals" Authors's response: corrected.

p. 3, l. 16: "how" instead of "the how"; Authors's response: corrected.

p. 4, l. 27: should be (Dickinson et al., 1993); Authors's response: corrected.

p. 5, l. 32-33: please give the references in parentheses; Authors's response: corrected.

p. 6, l. 7: should be (CH4); Authors's response: corrected.

p.7, l. 26: should be "Therefore". In this sentence please remove "respectively"; Authors's response: corrected.

p. 7, l. 29: should be "In Fig. 1"; Authors's response: corrected.

p. 7, l. 31: should be "for both cities"; Authors's response: corrected.

p. 8, l. 6: please use "7-8 pm" instead of "19-20 pm". Please correct throughout the MS; Authors's response: corrected.

p. 8, l. 26: should be "all aerosol types"; Authors's response: corrected.

p. 8, l. 27: please remove "as well"; Authors's response: corrected.

p. 8, l. 29: should be "for each component"; Authors's response: corrected.

p. 9, l. 5: should be "Po Valley in Northern Italy or in Southern Poland"; Authors's

response: corrected.

p. 9, l. 7: should be "Northern Italy". Please correct throughout the MS; Authors's response: corrected.

p. 9, l. 10: "is increasing" instead of "increased"; Authors's response: corrected.

p. 9, l. 13: "of wide" instead of wide"; Authors's response: corrected to There is a domain-wide ...

p. 10, l. 17: please add "respectively" in the end of the sentence; Authors's response: corrected.

p. 10, l. 21: "Fig. 7"; Authors's response: corrected.

p. 10, l. 21-23: please clarify the sentence starting with "In the chosen..."; Authors's response: The sentence was corrected to be more clear.

p. 12, l. 34-35: should be "...has also this large positive maximum, reaching..."; Authors's response: corrected.

p. 14, l. 33: "In areas..." instead of "Areas...". Authors's response: corrected.

References: Oleson, K.W., Bonan, G.B., Feddema, J., and Vertenstein, M. 2008. An urban parameterization for a global climate model. 2. Sensitivity to input parameters and the simulated urban heat island in offline simulations. J. Appl. Meteor. Clim. 47:1061- 1076.

---

## Author Comment (AC2) · 11 Sep 2018

Author response to the Referee #3's comments on manuscript "Impact of urban canopy meteorological forcing on aerosol concentrations" - acp-2018-415 by Peter Huszar et al.

We would like to thank to Referee #3 for the reviewing our manuscript and for all the comments, corrections and suggestions. We will consider all of them and our point-by-point responses follow.

Referee #3 comments:

Specific comment: In P10, L15 authors mentioned that "largest contribution is made by sulfates and nitrates being around 50 %, while sulfates dominate especially over

eastern European cities". What is the reason for the sulfate dominance over eastern C1 ACPD Interactive comment Printer-friendly version Discussion paper Europe?

Authors response: In the used emissions database (TNO MACC-III), emissions of SO2 over eastern European countries are higher than over western Europe while the opposite is true for the NOx emissions. As ammonium emissions are also slightly higher over western Europe, it is clear that over eastern Europe, sulphate ions will prefer to stabilize nitrate ions resulting in ammoium sulphate formation while over western Europe, the emissions ratios will favor the formation of ammonium nitrates (Schaap et a., 2004).

Changes in the manuscript: This detailed discussion was added to the manuscript to clarify the differences in contributions to the total PM2.5

Specific comment: It is not clear to me how the input data for building heights and street width is used in CLM4.5 on a 10 km × 10 km scale. What are the sources of this data?

Authors response: We provide the answer given to the other referee: As written in the manuscript, the urban morphology parameters are obtained from LandScan2004 global 2D data (Jackson et al., 2010) which defines 132 regional categories (the world is divided into 33 regions with similarities in urban characteristics and each category is subdivided into 4 subcategories representing different urban intensities - tall building district (TBD), high density (HD), medium density (MD), and low density (LD)). For each bottom category, average building heights (H), urban canyon height-to-width ratios (H:W), and fraction of pervious surface (e.g., vegetation), roof area, and impervious surfaces (e.g., roads and sidewalks) are defined, among other parameters. Jackson et al. provide all of the data sources from which these data were compiled. We checked the data for particular cities over the domain and they are within the range of the typical urban geometry represented by central European cities (see Huszar et al., 2014 for a few values representative for Prague, Czech Republic). Urban landunit within CLM4.5

is represented as fraction in percentages of three (of the four in Jackson et al.) urban intensity (HD, MD and LD). This gives a reasonable description of urban coverage even at 10 km resolution and even small cities well below 10 km in diameter are accounted for. Of course, within the model gridbox and within one urban intensity, urban parameters do not vary in space, however we consider this variation within the uncertainty range of other inputs like boundary conditions or physical parameterization etc.

Changes in the manuscript: We included some more detailed description (in Section 2.1.1) of how the urban parameters are obtained for the region in focus.

Specific comment: In P11, L19 authors commented "The SOA is an exception here, where concentrations are suppressed all over the domain peaking over urban areas up to -0.04 $\mu$gm$-3$". What is the reason for this exception?

Authors response: The SOA suppression here is the probably result of both larger removal of precursor semi-volatile species (SOA precursors) and the increased removal of SOA itself. These two processes concern the whole domain thus the SOA decreases everywhere. For inorganic aerosols, the latter process (concentration decrease) acts for the entire area, however, the primary species have different emission ratios across the domain thus turbulence impacts them with different magnitude and the competition between sulfates, nitrates and ammonium ions leads to different inorganic aerosol response.

Changes in the manuscript: This has been detailed in the revised manuscript.

Technical comments: Technical Comments: Please correct all the typing errors throughout the manuscript. I have listed some of them below:

Authors response: All typing errors have been corrected (some of them explicitly mentioned by the other reviewer)

References:

Oleson, K.W., Bonan, G.B., Feddema, J., and Vertenstein, M. 2008. An urban parameterization for a global climate model. 2. Sensitivity to input parameters and the simulated urban heat island in offline simulations. J. Appl. Meteor. Clim. 47:1061-1076.

Schaap, M., van Loon, M., ten Brink, H. M., Dentener, F. J., and Builtjes, P. J. H.: Secondary inorganic aerosol simulations for Europe with special attention to nitrate, Atmos. Chem. Phys., 4, 857–874, doi:10.5194/acp-4-857-2004, 2004.
* * *
**[ACPD](ACPD)**